# An Age-Period-Cohort Analysis of Biomarkers of Lifestyle-Related Diseases Using the National Health and Nutrition Survey in Japan, 1973–2018

**DOI:** 10.3390/ijerph17218159

**Published:** 2020-11-04

**Authors:** Tasuku Okui

**Affiliations:** Medical Information Center, Kyushu University Hospital, Fukuoka 812-8582, Japan; okui.tasuku.509@m.kyushu-u.ac.jp; Tel.: +81-092-642-5882

**Keywords:** cohort effect, Japan, systolic blood pressure, serum glucose, triglyceride, high-density lipoprotein cholesterol, body mass index

## Abstract

Studies of biomarkers of lifestyle-related diseases in Japanese cohorts are scarce. This study aimed to analyze trends in risk markers of lifestyle-related diseases using age-period-cohort (APC) analysis. Data on systolic blood pressure and BMI from 1973 to 2018 and serum glucose, triglyceride, and high-density lipoprotein cholesterol levels from 1989 to 2018 available from the National Health and Nutrition Survey were used. Values for each of the risk markers for each age, period, and cohort were estimated using APC analysis. For women, a decrease in all the risk markers of lifestyle-related diseases was observed in individuals born between the 1930s and approximately 1970. Therefore, female individuals born in approximately 1970 were considered to have the lowest risk of developing lifestyle-related and cardiovascular diseases. Meanwhile, the cohort effect on all the risk markers deteriorated for the younger cohorts, and changes in lifestyle behavior are needed for cohorts born more recently. For men, the trends in risk markers across the cohorts differed, and the relative risk of lifestyle-related diseases for each cohort differed according to disease. These results could help understand cohort-specific risks for lifestyle-related disease and enable identification of high-risk populations who could benefit from preventive measures.

## 1. Introduction

The increase in the number of Japanese patients with lifestyle-related diseases is of major concern [1,2]. In particular, obesity, diabetes, hypertension, and dyslipidemia are considered major lifestyle-related diseases in Japan. In this country, abdominal circumference, diabetes, hypertension, and dyslipidemia are used to diagnose metabolic syndrome [3], the incidence of which continues to increase. According to the Patients Survey in Japan [4], the estimated number of patients with hypertension increased from 7.49 (6.0% of the population) to 9.94 (8.0%) million from 1996 to 2017, diabetes increased from 2.18 (1.7%) to 3.29 (2.6%) million, and dyslipidemia increased from 0.96 (0.7%) to 2.21 (1.8%) million. The actual number of patients with these diseases is likely to be larger than the estimates, as many patients do not go to a hospital and remain undiagnosed. The increase in these diseases has also been associated with an increase in national medical expenses. According to national expense data for each disease [5], the expense for hypertension amounted to 1.79 trillion yen (approximately 17.1 billion USD) and the expense for diabetes amounted to 1.22 trillion yen (approximately 11.6 billion USD) in 2017. These two diseases account for approximately one-tenth of total national medical expenses. Additionally, patients with these diseases have a higher risk of cardiovascular diseases like ischemic heart disease, stroke, or cancer [6,7,8].

Although we can obtain the estimated number of patients for each type of disease, the prevalence is likely higher than the consultation rate since there are probably many untreated patients [9,10]. Therefore, these prevalence figures are somewhat uncertain. Although trends in mortality from cancer or cardiovascular disease have been often investigated using national data in Japan [11,12], similar studies investigating changes in the prevalence of the lifestyle-related diseases in specific years or birth cohorts in Japan are relatively limited [13]. Although the precise prevalence figures of the diseases are uncertain, by analyzing the trends in the average values of risk markers that are used for diagnoses of lifestyle-related diseases for the Japanese population, we might be able to deduce the trend in the risk for each type of disease.

The National Health and Nutrition Survey in Japan is a national survey that investigates data on the nutritional status, laboratory tests, and lifestyle of Japanese individuals. The survey investigates basic laboratory test values associated with diseases, such as body mass index (BMI), systolic blood pressure, and serum glucose over time, and investigations of the trends in average values have been conducted using the data [14,15,16,17,18]. However, the birth cohort effect on these test values has generally not been taken into account, and so the cohort effects on the laboratory test values are unknown. The mortality rates of cardiovascular disease and cancer vary considerably depending on birth cohort. The average laboratory values are also considered to be affected by the birth cohort effect, because patterns in lifestyle change according to cohort. By understanding the cohort effect on laboratory test values associated with lifestyle-related diseases, we may gain valuable insight into which cohorts need particular preventive measures or lifestyle changes.

In order to assess the cohort effect on these laboratory test values, Age-Period-Cohort (APC) analysis can be used. APC analysis is often used to examine trends in a disease incidence or mortality [19], and it identifies the changes in incidence or mortality by age, period, and cohort effects. APC analysis for laboratory test values such as BMI and systolic blood pressure has also been performed in other countries [20,21]. In this study, we analyzed trends in the physical status associated with lifestyle-related diseases in Japan using the National Health and Nutrition Survey data.

## 2. Materials and Methods

We used data from the National Health and Nutrition Survey in Japan [9], the aim of which is to investigate the status of nutrition intake, laboratory test values, and lifestyle patterns of Japanese people. About 300 districts were chosen from all over Japan using stratified random sampling, and all households in the districts were subject to the survey [22]. Although the number of subjects exceeded 20,000 people in earlier times, the number of subjects decreased throughout years [16,18,23]. In 2018, 3268 households were subject to the survey. In addition, the response rate to having a physical examination, such as blood pressure measurement, is said to be approximately 50–60% [16,18]. We used data on systolic blood pressure, serum glucose, triglyceride, high-density lipoprotein cholesterol (HDL-C), and body mass index (BMI) by age group and sex from 1973 to 2018 for the APC analysis. Data on serum glucose, triglyceride, and HDL-C were available from 1989. Systolic blood pressure was used for the diagnosis of hypertension, serum glucose for diabetes mellitus, HDL-C and triglyceride for dyslipidemia, and BMI for obesity [17,18,24]. Only data from subjects who had a blood test performed at least three hours after a meal were publicly available for serum glucose. Data from the National Health and Nutrition Survey have been used in many previous studies [14,15,16,17,18,25,26,27].

The National Health and Nutrition Survey in Japan is an annual cross-sectional survey, and the subjects are randomly sampled from the population of Japan every year. Therefore, subjects of a birth cohort are different depending on year. Age groups defined by 10-year increments from 20–29 to 60–69 years of age were available and used for analysis. Those who were 60–69 years old in 1973 (those born from 1904 to 1913) were the initial cohort, as they were the oldest birth cohort in the data set. Using a one-year shift in birth year, starting from the first cohort, the group aged 20–29 years in 2018 (those born in 1989–1998) were the last cohort. For the analysis of serum glucose, triglyceride, and HDL-C, those born from 1940 to 1949 were the initial cohorts.

We used the Bayesian APC model for analysis, which assumes that laboratory test values follow a normal distribution [20]. Let $y_{ij}$ be a laboratory test value for the age group i (1,…,I) in year  j (1,…,J). In the model, $y_{ij}$ are assumed to follow the following normal distribution whose mean is $\lambda_{ij}$,
$$y_{ij}\sim Normal\left(\lambda_{ij},\sigma^{2}\right),$$
$$\lambda_{ij}=\delta +\alpha_{i}+\beta_{j}+\gamma_{k}$$
δ is the intercept, $\alpha_{i}$ are the effects of age groups,$\;\beta_{j}$ are period effects,$\;\gamma_{k}\;\left(k=1,\ldots ,K\right)$ are cohort effects,$\;\mathrm{and}\;\sigma^{2}$ is the variance of $y_{ij}$. I, J, and K are the number of time points for age groups, periods, and cohorts, respectively. To identify each parameter, the sum of each effect was restricted to zero. A first-order random-walk was used as a prior for each effect. We used the Hamiltonian Monte Carlo method to estimate the parameters (http://mc-stan.org/). Next, estimates of each laboratory value were calculated for each age, period, and cohort. For example, the estimated value for age group i was calculated as $\;\hat{\delta }+{\hat{\alpha }}_{i}+{\hat{\beta }}_{median}+{\hat{\gamma }}_{median}$, where $\hat{\delta }\;and\;{\hat{\alpha }}_{i}$ were the estimated values for the intercept δ and the age effect $\alpha_{i}$, and ${\hat{\beta }}_{median}$ and ${\hat{\gamma }}_{median}$ were the medians of the estimated values of ${\hat{\beta }}_{median}$ and ${\hat{\gamma }}_{median}\;$ [28]. All statistical analyses were conducted using R v3.5.1 software (https://www.R-project.org/) (Open Source Software). In addition, the analysis of Bayesian APC model was conducted using rstan (https://cran.r-project.org/web/packages/rstan/rstan.pdf), and figures were created by using both ggplot2 (https://cran.r-project.org/web/packages/ggplot2/ggplot2.pdf) and cowplot (https://cran.r-project.org/web/packages/cowplot/cowplot.pdf).

## 3. Results

Figure 1 shows the yearly average values of systolic blood pressure, serum glucose, triglyceride, HDL-C, and BMI among men for the age groups from 1973 to 2017. Systolic blood pressure values decreased in all the age groups over the years. Serum glucose and triglyceride levels stayed relatively stable for most of the age groups. However, serum glucose level decreased suddenly in 1991 and 1992; the reason for the decrease is uncertain. HDL-C and BMI showed increasing trends with time.

Figure 2 shows the yearly average values of systolic blood pressure, serum glucose, triglyceride, HDL-C, and BMI for the age groups among women from 1973 to 2017. In contrast to men, serum glucose levels decreased from the mid-2000s in many of the age groups, and a decrease in BMI was observed in many of the age groups.

Table 1 shows the yearly age-adjusted average values of systolic blood pressure, serum glucose, triglyceride, HDL-C, and BMI among men and women from 1973 to 2017. Age-adjusted systolic blood pressure levels consistently decreased over the years among men and women. In addition, age-adjusted serum glucose levels decreased from 2005 for men and women, and those for HDL-C increased from 1989. Although the age-adjusted BMI increased for men, that of women decreased over the years.

Figure 3 shows the results of APC analysis of systolic blood pressure, serum glucose, triglyceride, HDL-C, and BMI for men. The estimated systolic blood pressure remained stable for cohorts born in approximately 1920. Serum glucose level remained stable in cohorts born between the 1930s and 1950s, showing a decreasing trend thereafter. Although triglyceride and HDL-C levels increased in cohorts born until approximately 1960, these tended to decrease thereafter. Although BMI showed a continuous increasing trend throughout the cohorts, the degree of increase decreased over the cohorts.

Figure 4 shows the results of APC analysis for systolic blood pressure, serum glucose, triglyceride, HDL-C, and BMI for women. Estimated systolic blood pressure continuously decreased over the cohorts, but began to increase in cohorts born around 1970. Serum glucose levels, triglyceride levels, and BMI decreased in cohorts born between 1930s and 1960s, but showed a decreasing trend thereafter. In contrast, an inverse trend was observed for HDL-C.

## 4. Discussion

Although an obvious decrease in the cohort effect was observed for women in cohorts born between approximately 1910 and 1970 for systolic blood pressure, the cohort effect for men remained stable for cohorts born after approximately 1920. Therefore, although period effects for systolic blood pressure continued to decrease for both sexes, the trends of the cohort effects differed to a large extent. Decrease of salt intake, increased use of antihypertensive agents, and reduced BMI are known to be associated with the general decline in systolic blood pressure levels in Japan [14,29]. In particular, the decrease of salt intake due to health promotion activities from the 1960s is considered to be a factor that contributed to the decline in systolic blood pressure and hemorrhagic stroke in Japan [30,31,32]. In addition, an increased treatment rate of hypertension by antihypertensive agents is also considered to have contributed to the general decrease in systolic blood pressure levels in Japan [16,32]. In fact, an increase in the treatment rate and a decrease in the salt intake were observed in the time periods [29,33], and thus considered to have affected the period effects. In addition, cohort effects on mortality rates for stroke and heart disease were shown to have decreased between cohorts born between approximately 1910 to 1960 for women [34], and these factors are considered to have also contributed to this phenomenon [29,30]. However, these factors appear to have contributed to the decline in the cohort effect on systolic blood pressure for men in a more limited way, and other factors may be more important among men. With the consumption of cigarettes in particular being shown to have increased after World War II [35], the cohort effect on the trend in smoking prevalence might be related to the trend in systolic blood pressure. On the other hand, for women, the cohort effect on systolic blood pressure began to increase in cohorts born in approximately 1970, and an increase in the cohort effect has also been observed for cardiovascular disease mortality rates [34]. The cohort effect on BMI was also shown to increase from the same time point for women, and risk factors such as reduced salt intake and BMI are considered to be related to this trend.

Regarding serum glucose levels, the trends in age, period, and cohort effects were relatively similar for both sexes, and estimated serum glucose levels decreased from 2006. Although consultation rates with regard to diabetes have been increasing for many age groups until 2017 for both the sexes [4], the trends in serum glucose levels were shown to differ. Analysis of the period effects demonstrated a decreasing trend. Another regional epidemiological study also showed no increase in the prevalence of type 1 or type 2 diabetes during the 2000s [36]. Thus, it is possible that the prevalence of diabetes did not increase in these periods. In addition, the cohort effect began to decrease in cohorts born in the 1930s and onwards for women, and in the 1950s and onwards for men. Although the cohort effect began to decrease in cohorts born in the 1950s for men, the effect did not increase from the 1930s. The trends in serum glucose levels and BMI were relatively similar for women. Although the reason for the decrease of the cohort effect on those born between approximately the 1930s and 1960s is uncertain, one possible explanation is the decrease of total energy intake accompanied by the reduction of carbohydrate intake [9,33]. Total energy intake was shown to be decreasing from 1995 in Japan [33], and there is a possibility that total energy intake decreased in the cohorts born between approximately 1930 and 1970. Total energy intake was shown to be associated with BMI for Japanese people, particularly for men [37], and the decrease in carbohydrate and fat intake is possibly reflected in the trend in serum glucose levels. It would be meaningful to conduct an APC analysis of the trend of total energy intake for both the sexes in the future. For men, trends in BMI and serum glucose levels were different in cohorts born after the 1950s, and multiple factors other than total energy intake, such as treatment rate of the diseases are considered also to be related to the cohort effect on blood sugar level.

The trends of the age effects for triglycerides were particularly different depending on sex for the age group older than the 40s. Previous studies have indicated that an increase in the prevalence of dyslipidemia starting from the 50’s age group is particularly evident for women in Japan [38]. Menopause leads to changes in the hormonal status, metabolism, and lipid profile [39,40]. Previous studies have reported that menopausal women had higher concentrations of triglycerides than pre-menopausal women with regular menstruation, whereas the concentration of HDL-C was significantly lower in menopausal women than in pre-menopausal women [39]. In addition, regarding triglyceride and HDL-C, the results for women were consistent. There was a decrease in estimated HDL-C levels and an increase in estimated triglyceride levels between cohorts born between the 1920s and 1960s. There were inverse trends observed from cohorts born after the 1960s. The possible reason for the trends is change in dietary habits because the trends were similar in blood sugar levels and BMI for women. Although it has been shown that the energy intake ratio from fat is continuously increasing in cohorts from around 1930 in both men and women [13], there is a possibility that fat intake decreased in the cohorts born between the 1920s and the 1960s because total energy intake possibly also decreases across the cohorts. For men, although HDL-C cholesterol levels increased in cohorts born between the 1920s and approximately 1960, triglyceride levels also increased in the same cohorts. The degree of change was larger for triglyceride, and the ratio of triglyceride to HDL-C level increased over the cohorts. The ratio of triglyceride to HDL-C has been shown to be associated with both cardiovascular disease risk and insulin resistance in overweight persons [41,42,43,44,45]. Increases in serum cholesterol levels are considered to be partly responsible for the increase in ischemic heart disease in Japan [46,47]. In a previous study, a surge in the cohort effect on ischemic heart disease (IHD) and cerebrovascular diseases was observed in male cohorts born in the mid-20th century [34], and this might be related to the increase in triglyceride over the cohorts. As the reason for the increase in estimated triglyceride, westernization of dietary habits and fat intake are considered to be a factor. On the other hand, the reason for the increase of HDL-C level over the analyzed periods for both men and women is still uncertain [26], and that for the trend of HDL-C level in men over the cohorts is also uncertain. Although it is known that HDL-C levels tend to decrease with increasing age [48], the values of the age groups overlapped for the result of HDL-C for men (Figure 1). Therefore, the reliability of HDL-C levels might be low for men.

Regarding BMI, period effects varied depending on sex. A higher incidence of overweight or obesity in the periods for men was also observed in an epidemiological study in Japan, while the mechanism remains uncertain [49]. The period effects on the major risk factors of obesity, such as smoking and alcohol drinking, significantly decreased in men in the period studied [50]. In addition, the prevalence of physical activity in women decreased in the periods covered, while it remained stable in men [50]. Thus, other factors like dietary habits are considered to be related to the result. The trend in the cohort effect on BMI has a lot in common with other laboratory values for women. Therefore, the trend in BMI is considered to be affected by dietary habits such as total energy intake, which are considered to be the main factors influencing BMI [13]. On the other hand, the cohort effect for men continuously increased in cohorts born between 1910 and 1970, and fat intake is a possible factor. In addition, the cohort effect showed relatively stable results for cohorts born around 1970, and the reason might be similar to that for triglyceride. Total energy intake decreased from the 1990s, whereas BMI continued to increase in the same period [12,33], and fat intake, but not total energy intake, is considered to be a possible factor in the increase of BMI over the years [27].

There are some limitations to this study. The number of subjects on whom laboratory tests were conducted varied depending on the laboratory test values and years, and ranged from approximately one thousand to six thousand [9]. Although the National Health and Nutrition survey in Japan is considered to be representative of the entire Japanese population [51], there were some cases in which laboratory test values changed considerably between adjacent time points. In addition, we analyzed the publicly available data, which were 10-year age groups. An analysis using finer age groups, such as age groups of 5-year increments, will provide a better picture of the cohort effects. Therefore, it is meaningful to perform a similar analysis using the data of nationwide epidemiological studies. As another limitation, APC analysis is a descriptive analytical method, and the precise reason for the changes in the cohort effects is still uncertain. APC analysis for dietary habits and lifestyle-behaviors is also needed for scrutinizing the trend in cohort effects. Even the reasons for the trend in the yearly obesity prevalence and serum cholesterol level are uncertain in Japan [17,27,52]. Many factors are considered to be associated with the trends in cohort effects on the laboratory test values, and by analyzing the trend in possible factors, we might better understand the reason for the changes in the cohort effects. On the other hand, the strength of this study is that the results are useful for understanding cohort-specific risks of lifestyle-related diseases and cardiovascular diseases.

## 5. Conclusions

To summarize the results of the cohort effect, an amelioration of all the risk markers for lifestyle-related diseases was observed in cohorts born between the 1930s and approximately 1970 for women. Therefore, cohorts born in approximately 1970 are considered to have the lowest risk of developing lifestyle-related diseases and cardiovascular diseases. On the other hand, the cohort effect on all risk markers deteriorated in more recent cohorts, and changes in lifestyle behavior should be encouraged in these recently born cohorts. For men, the increase in serum glucose and triglyceride levels ceased in the cohorts born in the 1950s. However, BMI did not show a decreasing trend thereafter, and systolic blood pressure stayed stable throughout the cohorts. Therefore, for men in Japan, the relative risk in each cohort for lifestyle-related diseases is considered to be different and depends on the individual diseases.

## Figures and Tables

**Figure 1 ijerph-17-08159-f001:**
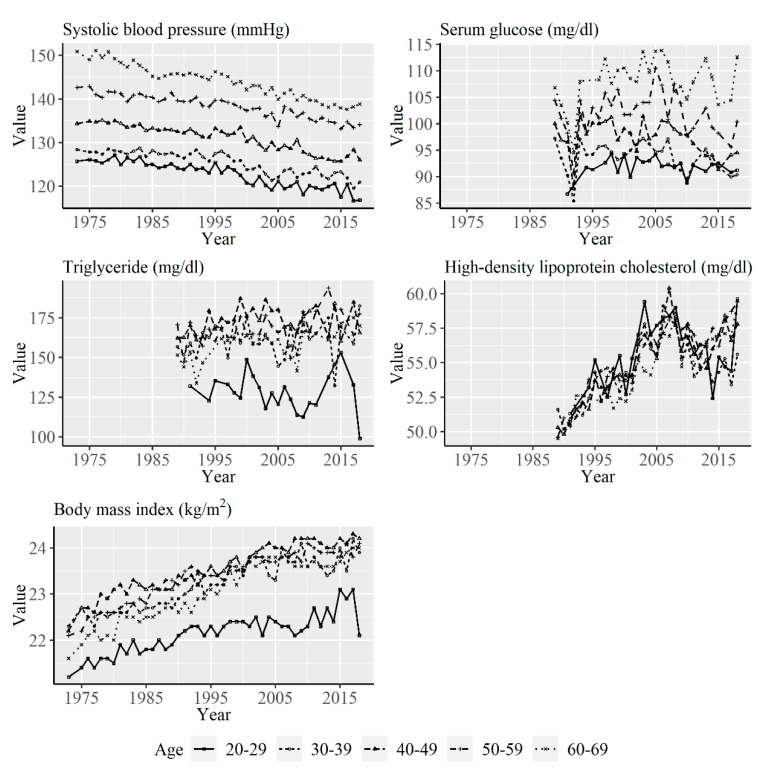
Yearly average values of systolic blood pressure, serum glucose, triglyceride, high-density lipoprotein cholesterol, and body mass index (BMI) according to age group among men from 1973 to 2017. Data on serum glucose, triglyceride, and high-density lipoprotein cholesterol were available from 1989.

**Figure 2 ijerph-17-08159-f002:**
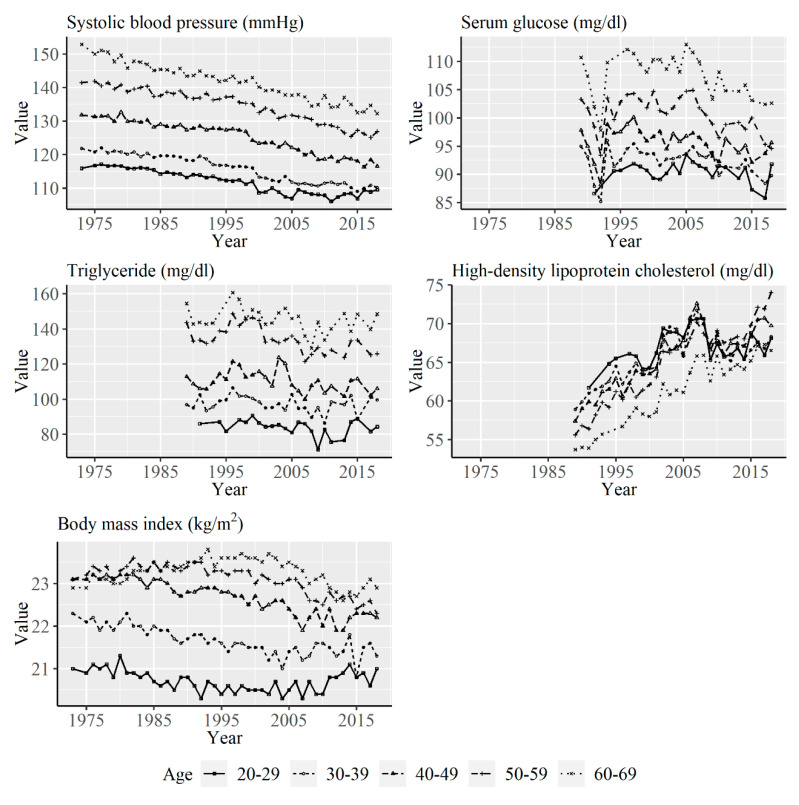
Yearly average values of systolic blood pressure, serum glucose, triglyceride, high-density lipoprotein cholesterol, and body mass index (BMI) according to age group among women from 1973 to 2017. Data on serum glucose, triglyceride, and high-density lipoprotein cholesterol were available from 1989.

**Figure 3 ijerph-17-08159-f003:**
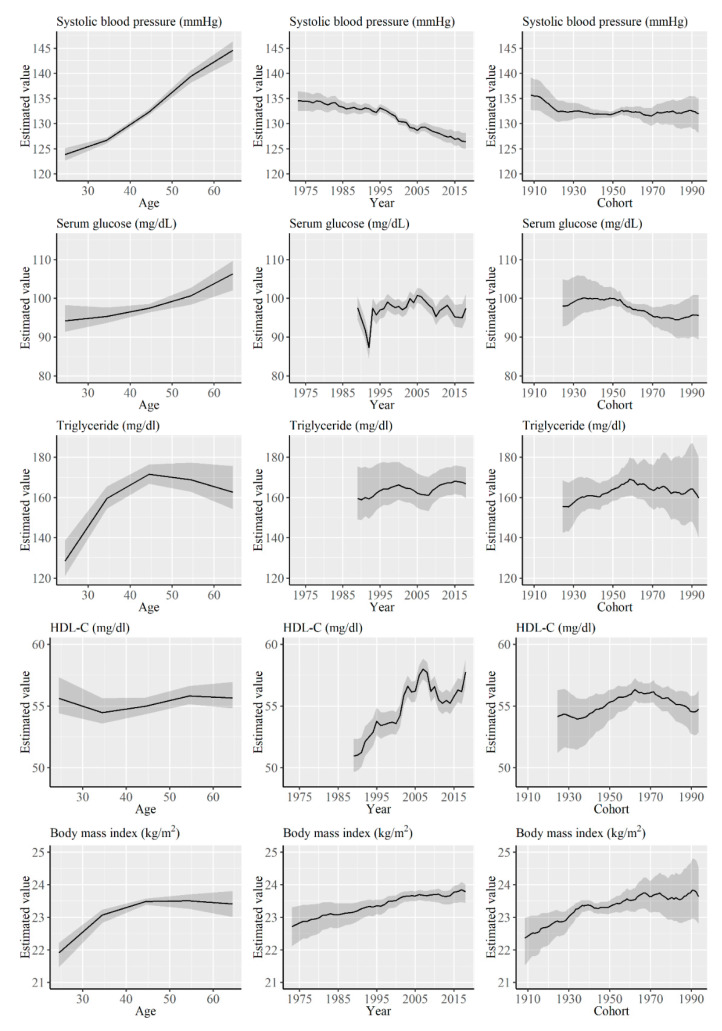
Results of the Age-Period-Cohort (APC) analysis of systolic blood pressure, serum glucose, triglyceride, high-density lipoprotein cholesterol, and body mass index for men. Solid lines signify estimates of each effect, and the shadings show 95% credible intervals of each estimates.

**Figure 4 ijerph-17-08159-f004:**
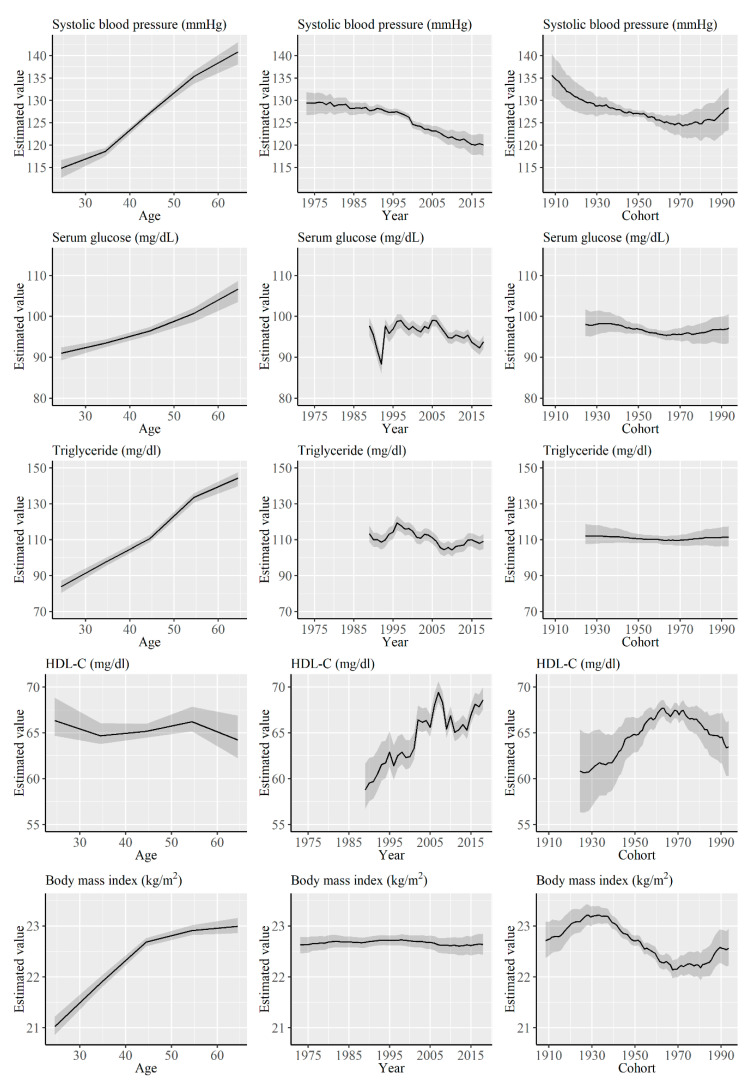
Results of the Age-Period-Cohort (APC) analysis of systolic blood pressure, serum glucose, triglyceride, high-density lipoprotein cholesterol, and body mass index for women. Solid lines signify estimates of each effect, and the shadings show 95% credible intervals of each estimates.

**Table 1 ijerph-17-08159-t001:** Age-standardized laboratory test values for men and women, 1973–2018.

	Men					Women				
	SBP *	SG †	TG ‡	HDL §	BMI ||	SBP *	SG †	TG ‡	HDL §	BMI ||
1973	137.2	_	_	_	21.9	134.1	_	_	_	22.6
1975	137.0	_	_	_	22.2	133.4	_	_	_	22.6
1976	136.9	_	_	_	22.3	133.6	_	_	_	22.7
1977	136.4	_	_	_	22.3	133.4	_	_	_	22.6
1978	137.1	_	_	_	22.4	132.1	_	_	_	22.7
1979	137.0	_	_	_	22.4	133.4	_	_	_	22.5
1980	136.2	_	_	_	22.4	131.2	_	_	_	22.7
1981	135.5	_	_	_	22.7	132.0	_	_	_	22.7
1982	136.2	_	_	_	22.6	131.7	_	_	_	22.7
1983	136.4	_	_	_	22.7	131.9	_	_	_	22.6
1984	135.0	_	_	_	22.6	130.2	_	_	_	22.6
1985	135.1	_	_	_	22.6	130.4	_	_	_	22.7
1986	134.4	_	_	_	22.7	130.5	_	_	_	22.6
1987	134.8	_	_	_	22.8	130.0	_	_	_	22.6
1988	135.3	_	_	_	22.8	130.6	_	_	_	22.5
1989	134.6	_	_	_	22.8	129.0	_	_	_	22.5
1990	134.2	_	_	_	22.9	129.2	_	_	_	22.6
1991	134.9	95.0	159.0	50.8	23.0	129.9	93.9	116.1	58.4	22.6
1992	134.8	_	_	_	23.0	129.3	_	_	_	22.6
1993	133.8	_	_	_	23.1	128.7	_	_	_	22.6
1994	133.2	_	_	_	23.0	128.3	_	_	_	22.5
1995	135.1	_	_	_	23.2	128.4	_	_	_	22.5
1996	134.6	_	_	_	23.1	128.6	_	_	_	22.5
1997	134.2	102.0	157.5	53.5	23.2	127.8	101.3	124.1	62.2	22.5
1998	133.5	100.8	161.6	53.6	23.4	127.5	99.7	120.5	62.6	22.5
1999	133.6	100.1	164.2	53.6	23.4	127.2	98.6	122.6	62.0	22.4
2000	131.6	100.4	166.5	53.4	23.3	124.8	99.6	121.4	62.3	22.4
2001	132.1	99.3	162.1	53.9	23.5	124.8	98.7	115.6	63.1	22.3
2002	132.2	99.5	159.6	56.0	23.6	124.8	97.9	115.0	66.7	22.3
2003	130.4	102.5	164.6	56.8	23.6	123.6	99.5	120.3	66.2	22.4
2004	130.8	100.5	161.2	56.1	23.6	123.9	98.7	119.3	66.4	22.2
2005	129.7	102.8	157.3	56.0	23.5	123.2	100.9	116.8	65.6	22.3
2006	131.2	102.4	155.8	57.9	23.7	123.3	100.8	115.0	68.4	22.3
2007	131.2	101.1	156.0	58.5	23.6	122.7	99.2	109.2	70.2	22.0
2008	130.6	99.9	150.1	58.2	23.6	122.0	97.7	108.7	68.9	22.1
2009	130.1	99.1	163.8	55.9	23.6	121.1	96.1	112.9	65.7	22.1
2010	129.8	96.4	161.7	57.1	23.6	121.9	96.0	108.0	67.7	22.0
2011	129.6	98.0	165.1	55.5	23.7	121.0	97.0	112.2	65.7	22.2
2012	129.2	_	_	55.2	23.6	120.7	_	_	66.2	22.0
2013	128.8	99.6	167.1	55.7	23.6	121.5	96.1	111.7	66.8	22.0
2014	129.1	97.8	163.2	55.5	23.6	120.5	97.0	116.2	66.1	22.2
2015	128.2	95.7	167.4	56.1	23.8	119.7	95.1	116.4	67.7	21.9
2016	128.9	_	_	56.9	23.8	119.7	_	_	69.1	22.1
2017	128.1	95.4	165.5	56.2	23.9	120.5	93.7	111.7	68.7	22.1
2018	128.1	98.4	161.7	58.4	23.8	119.8	95.3	114.8	69.4	22.0

* Systolic blood pressure (mmHg); † Serum glucose (mg/dl); ‡ Triglyceride (mg/dl); § High-density lipoprotein cholesterol (mg/dl); || Body mass index (kg/m^2^). The values of SG, TG, and HDL for some years after 1992 are missing because the data of some age groups were missing for those years. Laboratory tests for SG and TG were not conducted in 2012 and 2016.

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
