# Peer review of "An Age-Period-Cohort Analysis of Biomarkers of Lifestyle-Related Diseases Using the National Health and Nutrition Survey in Japan, 1973–2018"

_ijerph, 2020, doi:10.3390/ijerph17218159_

Round 1

Reviewer 1 Report

It is an innovative idea to use the APC model to investigate the age, period and cohort effects of biomarkers. I have several comments.

Q1. The authors stated in the method that “Next, estimates of each laboratory value were calculated for each age, period, and cohort. For example, the estimated value for age group ? was calculated as ?̂+?̂?, where ?̂ ??? ?̂? were the estimated values for the intercept ? and the age effect ??.” Here, the estimated age value should be the period-cohort adjusted age value, and also the age-cohort adjusted period value and the age-period adjusted cohort value. When estimating the adjusted value, the median effect of the other two should be included. Thus, the estimated age value should be adjusted age value = ?̂ + αi + βm + γm. Here are two references for authors to consider (https://jamanetwork.com/journals/jamanetworkopen/fullarticle/2749783; https://academic.oup.com/ije/article/47/1/81/3978809).

Q2. A 10-year age group is too wide and may miss some important information. If more detailed data are available, authors may use a smaller age range.

Reviewer 2 Report

Manuscript IJERPH 895262 v1. 25-10-2020

The study by Dr. Tasuku Okui presents cohort analysis of lifestyle-related disease markers in the Japanese population between 1973 (or 1989) until 2018. This research identifies females born between the 30s to 70s as with the lowest probability of developing lifestyle-related diseases, while men present much higher heterogeneity across years and diseases. The manuscript presents valuable data but needs some revisions before publication. These are detailed below and by manuscript section.

  1. Introduction. First paragraph.

                Reference number [3] should be placed before the comma.

                For non-Japanese readers, the data presented here could be difficult to interpret. Could the author provide approximate conversions from Yens to USD ? i.e. : 1.79 trillion yen (x USD). Besides, could the epidemiologic data also present % of Japanese population (as of 2020) ? i.e. : hypertension was found to increase from 7.49 (x %) to 9.94 (x %) million.

  1. Methods. Last paragraph.

                Please note that the latest R version is 4.0.3. (This is just a comment for the author, which does not need to be edited in the manuscript.)

                Can the author please mention which R packages were used for the data analyses ?

  1. Results.

                I cannot find any section (3 ??) on the results…? Did the author forget these paragraphs ? Are the Figures not explained in the result section ? The explanations seem to start at Figure 3…. Please amend.

                Figure 1. This Figure is very interesting and needs to be commented/explained in details. Indeed, it seems obvious that systolic BP decreased both with time and with birth year. This should be commented. The abscissa reads “202” instead of 2020.  Moreover, serum glucose presented a huge drop in 1991 (or 1992). Can the author explain why ?

Figure 1. All of these data should be checked thoroughly. Indeed, from the method section, the author stated that “the group aged 20–29 years in 2018 (those born in 1989–1998) were the last cohort”. However, a major concern is that data in Figure 1 is drawn for the 20-29 years old beginning in 1973 (at which point the subjects were not born !). It appears mathematically wrong, unless some explanations are missing. Similarly, the group aged 60-69 years old in 1973 would range from 105-114 years old in 2018, which means that a significant number of subjects would be deceased. The author should mention such a bias in the data set and/or in the data analysis.

Figure 2. Similar comments. Please see comments on Figure 1 (above).

Table 1. The legend does not indicate the unit for HDL. Please amend accordingly. Please also indicate that some data are missing in 2012 and 2016.

Figures 3 and 4. Similar comment than Figure 1 regarding the significant drop of serum glucose levels around 1991-1992, in both men and women.

Legends of Figures 3 and 4. Please change the word “credible” to “confidence” (i.e.: the 95% confidence interval).

  1. Discussion.

                Please revise the grammar in the sentence “Decreased salt intake due to health promotion activities from approximately the 1960s being considered to be a factor contributing to the decline in systolic blood pressure and hemorrhagic stroke in Japan [29-31].”, as it is difficult to read/understand at present.

  1. Conclusion.

                Please amend the following typographical error: “and changes in lifestyle behavior should be encouraged in the these relatively recently born cohorts.”.

Round 2

Reviewer 1 Report

The author mainly discussed the cohort effect, and left the age and period effect not discussed. Although the author mainly focused on cohort effect, a brief discussion of the age and period effect may help. Additionally, the results of age and period effect also showed some interesting findings. The author may need to discuss them. 

Author Response

Dear the referees,

Thank you for the kind comments. We reply to the comments for the manuscript `’ An age-period-cohort analysis of biomarkers of lifestyle-related diseases using the National Health and Nutrition Survey in Japan, 1973–2018 ‘’. 

  • To the comment “The author mainly discussed the cohort effect, and left the age and period effect not discussed. Although the author mainly focused on cohort effect, a brief discussion of the age and period effect may help. Additionally, the results of age and period effect also showed some interesting findings. The author may need to discuss them.  ‘’

→ We additionally noted about the age and period effects for each laboratory value in the Discussion.(Red colored parts in the Discussion) Also, the sentences were proofread by an English editor.

Reviewer 2 Report

Dear Author,

Thank you for revising your manuscript.

You have taken into account all of my concerns and revised adequately your manuscript. Therefore, I feel confident for approving its revised version.

I wish you the best with your ongoing research.

Author Response

Dear the referee,

Thank you for the comments.